# Glucocorticoid Receptor Gene (*NR3C1*) Polymorphisms and Metabolic Syndrome: Insights from the Mennonite Population

**DOI:** 10.3390/genes14091805

**Published:** 2023-09-15

**Authors:** Kathleen Liedtke Kolb, Ana Luiza Sprotte Mira, Eduardo Delabio Auer, Isabela Dall’Oglio Bucco, Carla Eduarda de Lima e Silva, Priscila Ianzen dos Santos, Valéria Bumiller-Bini Hoch, Luana Caroline Oliveira, Aline Borsato Hauser, Jennifer Elisabeth Hundt, Alan R. Shuldiner, Fabiana Leão Lopes, Teide-Jens Boysen, Andre Franke, Luis Felipe Ribeiro Pinto, Sheila Coelho Soares-Lima, Gabriela Canalli Kretzschmar, Angelica Beate Winter Boldt

**Affiliations:** 1Laboratory of Human Molecular Genetics, Department of Genetics, Federal University of Paraná (UFPR), Centro Politécnico, Jardim das Américas, Curitiba 81531-990, PR, Brazil; kathleenkolb@ufpr.br (K.L.K.); ana.mira@ufpr.br (A.L.S.M.); eduardoauer.biomedicina@gmail.com (E.D.A.); isabelabucco@gmail.com (I.D.B.); carladelimaesilva@gmail.com (C.E.d.L.e.S.); pri@ufpr.br (P.I.d.S.); valeriabumiller@gmail.com (V.B.-B.H.); luanacaroline.oliveira90@gmail.com (L.C.O.); gabriela.canalli@gmail.com (G.C.K.); 2Postgraduate Program in Genetics, Department of Genetics, Federal University of Paraná (UFPR), Centro Politécnico, Jardim das Américas, Curitiba 81531-990, PR, Brazil; 3Postgraduate Program in Internal Medicine, Medical Clinic Department, UFPR, Rua General Carneiro, 181, 11th Floor, Alto da Glória, Curitiba 80210-170, PR, Brazil; 4Laboratory School of Clinical Analysis, Department of Pharmacy, Federal University of Paraná (UFPR), Av. Pref. Lothário Meissner, 632, Jardim Botânico, Curitiba 80210-170, PR, Brazil; alinehauser@ufpr.br; 5Lübeck Institute of Experimental Dermatology, University of Lübeck, Ratzeburger Allee, 160, Haus 32, 23562 Lübeck, Germany; jennifer.hundt@uksh.de; 6Regeneron Genetics Center, 777 Old Saw Mill River Road, Tarrytown, NY 10591, USA; alan.shuldiner@regeneron.com; 7Human Genetics Branch, National Institute of Mental Health, 35 Convent Drive, Bethesda, MD 20892, USA; lopes.fabiana@gmail.com; 8Institute of Psychiatry, Federal University Rio de Janeiro, Av. Venceslau Brás, 71, Rio de Janeiro 22290-140, RJ, Brazil; 9Institute of Clinical Molecular Biology (IKMB), Christian-Albrechts-University of Kiel, 24105 Kiel, Germany; t.j.boysen@ikmb.uni-kiel.de (T.-J.B.); a.franke@mucosa.de (A.F.); 10Brazilian National Cancer Institute, Rua André Cavalcanti, 37, Rio de Janeiro 20231-050, RJ, Brazil; lfrpinto@inca.gov.br (L.F.R.P.); sheilacoelho@gmail.com (S.C.S.-L.); 11Faculdades Pequeno Príncipe, Av. Iguaçu, 333, Curitiba 80230-020, PR, Brazil; 12Instituto de Pesquisa Pelé Pequeno Príncipe, Av. Silva Jardim, 1632, Curitiba 80250-060, PR, Brazil

**Keywords:** *NR3C1*, HPA, glucocorticoid receptor, haplotype, anabaptist population, methylation, epigenetics

## Abstract

The regulation of the hypothalamic-pituitary-adrenal (HPA) axis is associated with polymorphisms and the methylation degree of the glucocorticoid receptor gene (*NR3C1*) and is potentially involved in the development of metabolic syndrome (MetS). In order to evaluate the association between MetS with the polymorphisms, methylation, and gene expression of the *NR3C1* in the genetically isolated Brazilian Mennonite population, we genotyped 20 *NR3C1* polymorphisms in 74 affected (MetS) and 138 unaffected individuals without affected first-degree relatives (Co), using exome sequencing, as well as five variants from non-exonic regions, in 70 MetS and 166 Co, using mass spectrometry. The methylation levels of 11 1F CpG sites were quantified using pyrosequencing (66 MetS and 141 Co), and the *NR3C1* expression was evaluated via RT-qPCR (14 MetS and 25 Co). Age, physical activity, and family environment during childhood were associated with MetS. Susceptibility to MetS, independent of these factors, was associated with homozygosity for *rs10482605*C* (OR = 4.74, *p*corr = 0.024) and the haplotype containing *TTCGTTGATT* (*rs3806855*T_ rs3806854*T_rs10482605*C_rs10482614*G_rs6188*T_rs258813*T_rs33944801*G_rs34176759*A_rs17209258*T_rs6196*T*, OR = 4.74, *p*corr = 0.048), as well as for the *CCT* haplotype (*rs41423247*C_ rs6877893*C_rs258763*T*), OR = 6.02, *p*corr = 0.030), but not to the differences in methylation or gene expression. Thus, *NR3C1* polymorphisms seem to modulate the susceptibility to MetS in Mennonites, independently of lifestyle and early childhood events, and their role seems to be unrelated to DNA methylation and gene expression.

## 1. Introduction

Metabolic syndrome (MetS) is a complex disease and a steeply rising cause of morbidity and death worldwide, increasing by more than two times the risk of mortality from cardiovascular diseases [1], which are the world’s leading cause of death [2]. Due to the intricate gene–environmental interactions, the common cause for MetS components, such as central adiposity, systemic arterial hypertension, insulin resistance, and dyslipidemia, is still unclear. The deregulated activity of the hypothalamic–pituitary–adrenal (HPA) axis is an emerging explanation, due to its influence on glucose and lipid metabolism, as well as anti-inflammatory and immune reactions [3]. The glucocorticoid receptor (GR) is responsible for mediating the negative feedback of the HPA axis, driven by cortisol levels [4]. To uncover its role in MetS development, higher environmental and genetic homogeneity, similar to the levels in animal models and isolated human populations, are desirable. The investigation of genetically isolated populations sharing the environment and demographic history for several generations benefits from the reduced variability and similar long-term exposure to confounding risk factors. The Mennonite population presents almost 500 years of isolation and three historical demographic bottlenecks (promoted by migrations mostly due to political–religious persecution), which increases the frequencies of uncommon alleles and allows for the use of smaller sample sizes for the identification of the loci associated with phenotypes [5,6], providing a unique opportunity to investigate the association of MetS with several parameters, like age, diet, physical activity, and paternal warmth in childhood.

Stress activates the sympathetic autonomic nervous system, followed by the HPA axis, which triggers the release of corticotropin-releasing hormone (CRH) and arginine vasopressin (AVP) by the paraventricular nucleus of the hypothalamus. In turn, they induce the pituitary gland to secrete adrenocorticotropic hormone (ACTH), which stimulates the adrenal cortex to secrete glucocorticoids (such as cortisol) into the bloodstream. Glucocorticoids bind GRs in the pituitary gland, hypothalamus, and hippocampus to regulate the production of CRH and ACTH, stabilizing their circulating levels [7]. Upon binding to glucocorticoids, GRs associate with co-chaperones and translocate to the nucleus, where they bind to glucocorticoid-responsive elements (GREs) and activate the expression of genes related to metabolic processes and immunity. Simultaneously, they interact with transcription factors, such as nuclear factor-kB (NF-kB) and activating protein-1 (AP-1), to reduce the expression of proinflammatory genes [3,4,8,9]. The deregulation of the HPA axis has indeed been related to MetS development, which is characterized by abdominal obesity, high triglycerides, low high density lipoproteins (HDL) cholesterol, arterial hypertension, and high fasting glucose [10]. MetS presents an interface with genetic and environmental factors, with physical inactivity and excessive caloric intake standing out among the latter [11]. The HPA axis in patients with MetS has reduced sensitivity to GR-mediated negative feedback, highlighting the potential role of the dysregulation of this system in the disease [12], with excess cortisol being associated with hypertension, visceral obesity, and diabetes/resistance to insulin, in addition to mood and cognition disorders [13].

The glucocorticoid receptor gene (*NR3C1)* (nuclear receptor subfamily 3 group C member 1) encodes GR and is of particular interest in investigating the association between genetic factors and diseases that involve HPA axis imbalance [14,15,16]. This gene is located on the reverse strand of chromosome 5q31.3 (GRCh38.p13:chr5:143,277,931-143,435,512), contains eight coding exons (2 to 9) and nine non-coding exons (1A, 1I, 1D, 1J, 1E, 1B, 1F, 1C and 1H), located in the proximal regulatory region (5’UTR), and each one has its own promoter region. There are two main isoforms resulting from the alternative splicing of *NR3C1* pre-mRNA: GRα and GRβ. GRα is the most frequent isoform, while GRβ has about 1% of the expression of the GRα variant and acts as its dominant inhibitor [17]. *NR3C1* is epigenetically regulated by DNA methylation, presenting a CpG island in its 5’UTR region that comprises multiple exons “1”, except for exons 1A and 1I [18,19]. Differential 1F exon methylation has been associated with bipolar disorder, borderline disorder, depression, and post-traumatic stress disorder, which, in turn, are also related to HPA axis dysregulation [7,16,20,21]. These DNA methylation alterations have been widely attributed to the influence of stressors during critical periods, mainly in childhood, such as child maltreatment (neglect; exposure to violence by an intimate partner or to physical, emotional, and sexual abuses) and pain-related stress [16,22,23]. In rats, it has been shown that enhanced pup licking and grooming and arched-back nursing by mothers alters the offspring epigenome in exon 17 of *NR3C1* (homologous to exon 1F in humans) [24], and maternal caregiving behaviors in humans reduced early stress exposure [23].

*NR3C1* promoter methylation is also associated with lower GR expression, reducing sensitivity to the HPA axis negative feedback [25]. While studies on *NR3C1* methylation focus on psychopathologies, with scarce studies on MetS or its components, several polymorphisms have already been associated with them, with emphasis on the single-nucleotide polymorphisms (SNPs) rs56149945 (p.N363S), rs41423247 (*Bcl*I), rs6189/6190 (*ER22/23EK*), rs10052957 (*TthIII*I), and rs6198 (GR-9β), which seem to modulate the sensitivity to glucocorticoids [3,26]. However, many of these studies lack expression analysis, and the functional impacts of these SNPs are still poorly understood. 

In this work, we gained important insights into the association of GR genetic polymorphisms/haplotypes, DNA methylation markers, and gene expression in the leukocytes of South Brazilian Mennonites with MetS, where we investigated 25 *NR3C1* polymorphisms, the methylation levels of 11 CpGs mapped to the 1F region, and *NR3C1* mRNA levels.

Exome sequencing enabled the investigation of 20 *NR3C1* polymorphisms, not yet investigated in the literature in relation to MetS or its risk factors, giving rise to a new possible genetic variation role in MetS etiology. The other five *NR3C1* SNPs were selected based on their association with factors related to MetS, e.g., in the European population, rs6877893 was associated with reduced waist circumference adjusted for body mass index (BMI) [27] and *rs258763*T*, *rs7701443*G*, and *rs72802813*A* alleles were associated with reduced hip circumference adjusted for BMI [28]. Furthermore, *rs41423247*G* polymorphism was associated with hypersensitivity to glucocorticoids [29], higher blood pressure, and insulin and glucose levels in obese northern Indians [30], as well as increased BMI, waist circumference, and systolic blood pressure in congenital adrenal hyperplasia Brazilian patients [31]. In addition, *rs41423247*G/G* homozygotes exhibited an increased risk of developing MetS in the Turkish [32] and Chinese [33] populations and presented higher BMI, body weight, abdominal obesity, fasting glucose, and insulin in Swedish men [34].

## 2. Materials and Methods

### 2.1. Research Participants

This research was approved in two instances by the Ethics Committee of the UFPR Health Sciences Sector (CAAE 55528222.9.0000.0102 and 55297916.6.0000.0102). After informed consent, we collected peripheral blood from 349 Mennonite volunteers from three southern Brazilian communities, from 2016 to 2022: 126 from the urban community of Curitiba (CWB, PR) and 84 and 139 from the two rural settlements of Colônia Witmarsum (CWI; Palmeira, PR) and Colônia Nova (CON; Aceguá, RS), respectively. All participants had their biometric parameters measured and were interviewed with a questionnaire based on the National Healthy Survey [35] to evaluate health and lifestyle conditions, as well as familial disease aggregation. Inclusion criteria were the Mennonite origin for at least one of the parents (sharing a common migratory route from the Netherlands to Poland, then to Ukraine, and from there again to Germany and later to Brazil or Paraguay); more than 12 years of age; and capacity to understand and answer the questions of the questionnaire. Individuals with MetS were classified based on the modified Joint Interim Statement (JIS) [10] and should have at least three of the criteria listed in Table 1. Exclusion criteria were controls with first-degree ascending relatives with patients. The demographics and epidemiologic data of the participants are shown in Table 2. In total, 112 individuals presented MetS, and 237 were considered healthy controls. As of August 2021, blood samples were also collected in PAXgene Blood RNA tubes (Becton Dickinson, Vaud, Switzerland). 

### 2.2. NR3C1 Genotyping

Peripheral blood was collected in tubes containing EDTA, and DNA was extracted from the buffy coat, using a Wizard^®^ Genomic DNA Purification Kit (Promega, Fitchburg, WI, USA), according to the manufacturer’s instructions. We selected 05 single-nucleotide polymorphisms (SNPs) based on previous studies showing their association with some of the MetS criteria and a minor allele frequency (MAF) higher than 0.10 in the European population (Utah, USA) [36]. We evaluated the following SNPs: rs72802813 (5:143423467, in intron 1, *ENST00000343796.6:c.-14+11065C>t*), rs7701443 (5:143413085, in intron 1, *ENST00000343796.6:c.-13-12233T>c*), rs41423247 (*BclI*; 5:143399010, in intron 2, *ENST00000343796.6:c.1184+646C>g*), rs6877893 (5:143347628, in intron 2, *ENST00000343796.6:c.1185-33460C>t*), and rs258763 (5:143272796, in intergenic region *ARHGAP26, NR3C1*, *NC_000005.10:g.143272796T>a*). Up to 166 controls and 70 patients were genotyped using the iPLEX MassARRAY Platform (Agena Bioscience, San Diego, CA, USA).

In addition, 212 exomes from Mennonites were previously generated by our research group with >30× coverage (Illumina HiSeq) (Illumina, San Diego, CA, USA). From these data, we performed screening, selecting the variants located in the *NR3C1* gene after filtering with the VEP (Variant Effect Predictor) tool—Ensembl! [37], so we evaluated 20 *NR3C1* polymorphisms (Table 3) in 74 additional individuals with MetS and compared them with 138 controls. 

### 2.3. Quantitative Pyrosequencing DNA Methylation Assay

We evaluated DNA methylation in the buffy coat in up to 141 controls and 66 individuals with MetS, investigating 11 CpG sites mapped to a CpG island within the *NR3C1* 1F region, whose differential methylation was previously associated with HPA axis dysregulation. CpG sites were selected according to the standard numbering sites described by Palma-Gudiel et al. [20] and position based on the GRCh38 genomic sequence: 35 (5:143404075), 36 (5:143404073), 37 (5:143404063), 38 (5:143404057), 39 (5:143404043), 40 (5:143404020), 43 (5:143403983), 44 (5:143403976), 45 (5:143403973), 46 (5:143403967), and 47 (5:143403964) (Figure 1). The bisulfite conversion of genomic DNA was performed using an EZ DNA Methylation Kit (Zymo Research, Irvine, CA, USA), following the manufacturer’s recommendations. Amplicons were generated using primers designed using the PyroMark Assay Design Software v. 2.0.1.15 (Qiagen, Hilden, Germany) (amplification primers and sequencing primer 1) or manually designed (sequencing primers 2 and 3) according to the reported recommendations for pyrosequencing primers [38]. Primers are shown in Appendix A.

Polymerase chain reaction (PCR) tests were performed in a final volume of 50 µL, containing 25 ng of bisulfite-treated genomic DNA, 0.2 mM of each dNTP and 0.2 µM of each PCR primer, 1 U/rxn Taq Platinum (Invitrogen Life Technologies, CA, USA), and 1× CoralLoad PCR buffer (Qiagen, Hilden, Germany). Thermal cycling started at 95 °C for 15 min, followed by 50 cycles, each starting at 95 °C for 40 s; an annealing step at 57 °C for 40 s; and ending at 72 °C (extension step) for 40 s. To confirm amplification and amplicon size, we submitted the amplified fragments to an electrophoretic run on 1% agarose gel, stained with UniSafe Dye^®^ 20,000× (Uniscience do Brasil, SP, Brazil). Subsequently, 40 μL of PCR products were immobilized on streptavidin-coated sepharose beads (GE Healthcare, IL, USA). Pyrosequencing was performed using a PSQ 96 ID Pyrosequencer (Qiagen, Hilden, Germany) with a PyroMark Gold Q96 Reagent Kit (Qiagen, Hilden, Germany), and the methylation percentage for each CpG site was automatically generated using the PyroMark Q96 software v. 2.5.8 (Qiagen, Hilden, Germany) with standard quality control settings. 

Exons are represented by boxes and introns are indicated by lines. *NR3C1* is composed of eight coding exons, numbered 2–9, and nine first non-coding exons. The first two boxes represent exons 1A and 1I. Exons 1D, 1J, 1E, 1B, 1F, 1C, and 1H are located within a CpG island, represented by a single box. A fragment of the 1F sequence is displayed, with the red color indicating the investigated CpG sites. The positions of SNPs comprising the investigated haplotypes are depicted. Locations of SNPs too close to each other are indicated by one single arrow. 

### 2.4. RNA Extraction, cDNA Synthesis, and RT-qPCR

Total RNA was isolated from the buffy coat with Quick-RNA™ Miniprep Kit Zymo: R1054 (Zymo Research, Irvine, CA, USA), adapted to PAXgene tubes, and reverse-transcribed with a High-Capacity cDNA Reverse Transcription Kit (Applied Biosystems, San Francisco, CA, USA). Gene expression levels were quantified with qPCR using TaqMan probes (Thermo Fisher Scientific, Waltham, MA, USA) for the *NR3C1* (assay ID Hs00230818_m1) and glyceraldehyde-3-phosphate dehydrogenase (*GAPDH*) (assay ID Hs03929097_g1)*,* as an endogenous control gene. All assays were performed in triplicates, and *NR3C1* relative mRNA levels were normalized via *GAPDH* mRNA expression. RT-qPCR was performed using ViiA 7 Real-Time PCR System (Thermo Fisher Scientific, Waltham, MA, USA). Ct values (threshold cycle) were calculated using the ViiA 7 Software v1.3 (Thermo Fisher Scientific, Waltham, MA, USA), and gene expression was calculated using the comparative Ct method 2^−ΔΔCt^ [39]. In this phase of the study, we evaluated 14 MetS individuals and 25 controls.

### 2.5. Statistical and Bioinformatic Analysis

To check for the false discovery rate, the *p*-value was corrected for association tests as described below. The corrected *p*-value is hereinafter referred to as *p*corr and was considered statistically significant when lower than 0.05 (*p*corr < 0.05).

Binary univariate logistic regression was used to establish independent variables, for which the association tests should be adjusted, using MetS phenotype as the dependent variable. The *p*-value was adjusted using the Benjamini and Hochberg [40] approach. Predictive variables that achieved *p*corr values lower than 0.20 were included in the binary multivariate logistic regression model, carried out using the backward method, in which variables with less significance were removed one at a time from the model until all the present variables were statistically significant. R Statistical Software v4.2.2 [41] was used.

Exome raw data were converted to the Variant Call Format (VCF) and aligned to the reference genome GRCh38/hg38, verifying the quality of the sequencing using the ForestQC software v. 1.1.5.7 [42].

We identified extended SNP haplotypes based on the haplotype block estimation method by Gabriel et al. [43] performed in Haploview 4.1 [44]. As inclusion criteria of variants to reconstruct the haplotypes, only SNPs with MAF higher than 0.10 in the studied Mennonite population were considered (Figure 1). We also used Haploview 4.1 to evaluate linkage disequilibrium (LD) (Appendix A). Phase information about SNP haplotypes was inferred using the ELB algorithm implemented in Arlequin v.3.5.2.2 [45]. Only haplotypes with a frequency higher than 0.10 in Mennonites were included in the genetic association tests.

We obtained allele, genotype, and SNP haplotype frequencies through direct counting and tested genetic associations within the dominant, recessive, and additive models, as well as the hypothesis of the Hardy–Weinberg equilibrium, with PLINK 1.9 software [46]. We further compared the polymorphism and haplotype distribution between the investigated groups using multivariate logistic regression, adjusted for the possible effects of independent variables (age, family environment in childhood, and moderate or vigorous physical activity), performed in R Statistical Software v4.2.2 [41]. The *p*-value was corrected using the Monte Carlo permutation method for SNP associations and Benjamini and Hochberg correction for haplotype associations. As an effect measure, we used the odds ratio (OR) with 95% confidence interval (CI).

The distributions of polymorphisms and haplotype frequencies between populations were compared using Fisher’s exact test with R Statistical Software v4.2.2. To obtain population polymorphism frequencies, we accessed gnomAD v3.1.2 https://gnomad.broadinstitute.org/ (accessed on 4 March 2023) [47] and ABraOM v2.1 (https://abraom.ib.usp.br/) (accessed on 4 March 2023) [48] databases. Haplotype frequencies in the European population (Utah, USA) were accessed in LD link v5.6.0 (https://ldlink.nci.nih.gov) (accessed on 9 May 2023) [49]. All frequencies were obtained considering the GRCh38 genome version.

*NR3C1* methylation levels and gene expression did not follow a normal distribution (tested with Shapiro–Wilk test and D’Agostin–Pearson test), so comparisons were performed with non-parametric tests (Mann–Whitney test and Kruskal–Wallis test), carried out using GraphPad Prism v.5.01 (GraphPad Software, San Diego, CA, USA). 

## 3. Results

The distribution of MetS did not differ between settlements (*p*corr = 0.78), sexes (*p*corr = 0.39), or urban (CWB) and rural (CWI and CON) environments (*p*corr > 0.99) in the univariate analysis. Higher age (OR = 1.05 [95%CI = 1.03–1.07], *p*corr < 0.001) and lower familiar warmth in infancy (OR = 1.59 [95%CI = 1.08–2.34], *p*corr = 0.019) were independently associated with MetS susceptibility. The family environment in childhood was reported as a simple answer to the question “How was your family environment during childhood?” with three possibilities: warm (lots of affection and hugs), moderate (disciplined), or cold (distant). On the other hand, daily moderate or vigorous physical activity for over 10 min was independently associated with MetS protection (OR = 0.44 [95%CI = 0.26–0.73], *p*corr = 0.003). Participants were asked if they perform daily moderate or vigorous activities for at least 10 min without interruption at work, for leisure, sports, exercise, or as part of their activities at home, in the yard, or any other activity that increases their breathing or heart rate, such as cycling, swimming, dancing, aerobics, running, playing sports, carrying weights, doing household chores around the house or yard such as sweeping or jobs such as stacking boxes, using a hoe, sledgehammer, etc. Waist circumference was the most frequent diagnostic parameter, which was observed in all individuals with MetS. Genotypes of controls and patients were distributed according to the predictions of the Hardy–Weinberg equilibrium.

### 3.1. NR3C1 Polymorphisms and Susceptibility to Metabolic Syndrome

Allele and haplotype frequencies are shown in Table 4 and Table 5. The homozygosis of the *rs10482605*C* allele was associated with increased MetS susceptibility (OR = 4.74 [95%CI = 1.11–20.29], *p*corr = 0.024). Most of the allele frequencies differed from those of the European non-Finish population (15 of 25 SNPs), Amish population (18 of 25 SNPs), and Brazilians (17 of 21 SNPs) (Appendix A). We identified three haplotype blocks reconstructed based on linkage disequilibrium between the SNPs and the nine haplotypes with a frequency higher than 0.10 (Appendix A and Appendix A). Homozygote individuals for the *CCT* haplotype (formed by rs41423247, rs6877893, and rs258763) showed a higher susceptibility to MetS (OR = 6.02 [95% CI = 1.41–25.62], *p*corr = 0.030). Homozygous individuals for the *TTCGTTGATT* haplotype (formed by rs3806855, rs3806854, rs10482605, rs10482614, rs6188, rs258813, rs33944801, rs34176759, rs17209258, and rs6196) also had an increased probability of developing MetS (OR = 4.74 [95% CI = 1.10–20.28], *p*corr = 0.048) (Table 5). Among the evaluated haplotypes, only *GCTATTGATC* frequency differed from the European population (Utah, USA) (*p* = 0.0002).

### 3.2. NR3C1 Methylation and Susceptibility to Metabolic Syndrome

The methylation profile of the CpG sites mapping to *NR3C1* 1F region did not show a significant association with MetS, neither individually nor when their median levels were considered (Table 6 and Appendix A). We observed that the region encompassing these CpG sites was mostly unmethylated (0% methylation) or presented very low methylation levels. 

### 3.3. NR3C1 mRNA Expression Levels

*NR3C1* mRNA expression levels were evaluated according to the genotypes for both haplotypes associated with MetS in the present study: *CCT* and *TTCGTTGATT*. The *rs10482605*C/C* genotype was also evaluated through this later analysis since all individuals with this genotype presented the *TTCGTTGATT* haplotype. None of these genotypes were associated with differential mRNA expression levels in any of the comparisons: (a) homozygote individuals relative to heterozygous individuals; (b) homozygote or heterozygote individuals relative to individuals with other genotypes; (c) homozygote individuals relative to individuals with other genotypes; and (d) homozygote individuals relative to individuals neither homozygote nor heterozygote. *NR3C1* mRNA expression levels were also compared between MetS individuals and controls, but no association was found (Figure 2).

## 4. Discussion

The investigation of *NR3C1* gene methylation has gained prominence due to its relationship with the regulation of the HPA axis, as well as its association with psychopathologies, and it is still poorly explored for MetS. In contrast, *NR3C1* polymorphisms are associated with the comorbidities that constitute MetS, but their association with MetS itself is still unclear. The *NR3C1* gene is highly expressed in the brain, and its main transcripts are similarly expressed in brain tissues and whole blood [50]. Thus, methylation and gene expression profiles in the buffy coat probably reflect those in the brain, and may also reflect alterations in the HPA axis [51]. In the present study, we investigated *NR3C1* polymorphisms, methylation patterns in CpGs mapped to the 1F region, and their expression in buffy coats from MetS subjects and controls of an isolated population. Interestingly, along with the risk factors known to be associated with MetS susceptibility, such as daily physical activity (protective) and higher age (risk), lower familiar warmth in infancy increased the odds for MetS in this population, reinforcing the possibility that epigenetic markers of HPA axis genes play a role in the response to childhood stress and MetS development. We found two *NR3C1* haplotypes and one SNP associated with MetS susceptibility but no differences in its methylation or gene expression levels in leukocytes. Differences in these parameters may rather be found in other tissues directly related to the HPA axis.

The minor allele frequencies of the analyzed SNPs reflect the possible influence of bottlenecks and/or founder effect in our investigated Mennonite population, since most of them differed from other populations, including the European non-Finish, Amish, and Brazilian populations. The SNPs that were associated with MetS susceptibility, alone or in haplotype, occur in regulatory regions with CpG islands, open chromatin histone marks, and transcription factor binding sites. Four SNPs are mapped to the CpG island (rs3806854, rs3806855, rs10482605, and rs10482614) located in the 5’UTR region. Considering the myeloid and lymphoid cell lineages, with the exception of rs258763, the other SNPs occur in regions with enhancers or histone marks associated with transcription activation such as H4K20me1, H3K4me1, H3K4me2, H3K4me3, H3K27ac, and/or H3K9ac, and rs10482605, rs10482614, rs3806854, and rs3806855 bind more than ten regulatory proteins [52]. Three of the polymorphisms have a potential for regulatory role interference, as the minor allele disrupts (rs10482614) or creates (rs258813 and rs3806854) CpG sites, with rs3806854 occurring in intron 1B and therefore potentially susceptible to methylation. In addition, seven of the SNPs were associated with differences in gene expression (eQTL—expression quantitative trait loci) in visceral adipose, brain, and/or cardiovascular tissue, as their minor allele was associated with lower expression in whole blood (rs3806854, rs3806855, and rs10482614), aorta (rs3806854, rs3806855, rs10482614, rs41423247, rs6188, and rs258813), adipose tissue (subcutaneous) (rs41423247), and brain cortex (rs6188 and rs258813). On the other hand, the minor allele of rs6877893 was associated with gene overexpression in the aorta, and rs3806854, rs3806855, and rs10482614 were associated with overexpression in the brain’s *substantia nigra* [50].

The minor alleles of *rs7701443*T>C* and *rs72802813*C>T*, which occur in the same haplotype, were associated with reduced BMI-adjusted hip circumference in Europeans [28]. Nevertheless, we did not find an association of these SNPs or their haplotype with MetS in the Mennonite population. 

The *rs10482605*C* allele is exclusively located within the *TTCGTTGATT* haplotype. As expected, homozygosity for both was associated with MetS susceptibility. The rs10482605 minor allele has been associated with an increased risk of developing stress-related disorders, such as depression [53]. This allele has been reported to be in linkage disequilibrium with *rs6198*G* (GR-9β variant), which disrupts an *ATTTA* motif within the 3′UTR of exon 9β, giving rise to a *GTTTA* sequence [53]. The *AUUUA* motifs are known to destabilize mRNA through recognition by RNA binding proteins in AU-rich elements (ARE) that enhance deadenylation and decay, recruiting mRNA degradation machinery [54,55,56,57,58]. AU-rich motifs may also enhance microRNA (miRNA) function in translational repression [59], (but none of the miRNAs whose action on the *NR3C1* mRNA was experimentally confirmed so far recognize the sequence containing rs6198) [60]. The disruption of the *AUUUA* motif is expected to stabilize the mRNA and increase GRβ protein expression, which is also associated with glucocorticoid resistance [32,61,62]. In fact, MetS patients overexpress GRβ in PBMCs, suggesting its involvement in glucocorticoid resistance and HPA dysregulation [12]. Interestingly, both isoforms diverge only in the C-terminal region, influencing translocation to the nucleus and transactivation of other genes. GRβ inhibits the transcriptional activity of GRα through yet poorly understood mechanisms, probably through the formation of inactive heterodimers with GRα, or competition for binding on GREs or binding with coactivators [63,64,65]. 

The rs41423247 polymorphism, or *Bcl*I, is an intronic SNP whose minor allele *G* has been associated with increased sensitivity to glucocorticoids [29,34]. Furthermore, *G/G* homozygosity has been associated with an increased risk of developing MetS [29,32,33,34]. In contrast, homozygosity for the minor allele was not observed among Brazilian MetS patients, who also presented reduced glucocorticoid sensitivity [12]. Our results partially agree with the Brazilian study, since we only found a MetS susceptibility association with homozygosity of a haplotype containing the rs41423247 major allele (*rs41423247*C/rs6877893*C/rs258763*T*) associated with glucocorticoid resistance. 

We did not find any difference in *NR3C1* global expression levels in individuals with or without homozygosity for the associated alleles (Figure 2). In accordance, Cao-Lei and collaborators [19] also reported no association between the minor allele of rs10482605 and exon 1C promoter activity, while Kumsta et al. [53] found reduced activity in two brain cell lines. Notably, we did not differentiate mRNA isoforms. The detection of subtle yet significant differences may have been possible with an increased sample size. Furthermore, the activity of alternative *NR3C1* promoters varies between cell lines in vitro [19], and the investigation of other relevant tissues for MetS might be useful. 

Small *NR3C1* methylation differences (<10%) have been identified as related to the development of disease phenotypes [25], with differences in methylation in the 1F region of the *NR3C1* gene being widely associated with psychopathologies, mainly correlated with early life adverse events or in childhood [16,21,23]. Changes in methylation patterns are more susceptible to stressful events experienced in these periods [23], also in animal models [24]. In our study, we indeed observed an association between MetS susceptibility and lower familial warmth in infancy, regardless of other factors. However, there were no methylation differences between the buffy coats of individuals with and without MetS in the CpGs evaluated for the 1F region, even though the expression of these genes is similar in blood and hypothalamus. Although no association studies of MetS with *NR3C1* differential methylation were found in the literature, variations in its methylation patterns have been observed for related comorbidities, such as cardiovascular diseases [22], subclinical arteriosclerosis (hypermethylation of the 1F promoter, in a study with monozygotic twins) [66], overweight (hypomethylation of the 1F region, CpGs 40 to 47) [51], unfavorable prognosis for coronary acute syndrome in individuals with depression (hypermethylation of exon 1F) [67], blood pressure (hypermethylation of 1F and 1H promoters associated with lower blood pressure) [68], and a positive association between methylation and glucose levels as well as insulin resistance [69]. Considering the multiple alternative first exons and their variability in tissue-specific expression, and that each of these exons has its own active promoter, DNA methylation and other epigenetic mechanisms should be evaluated in other regions. Also, measuring methylation patterns in cells other than leukocytes, especially those directly involved in the HPA axis, would be desirable.

## 5. Conclusions

With this study, we reinforce the potential association of *NR3C1* polymorphisms with MetS development, probably due to HPA axis dysregulation. Future research should evaluate haplotypes with rs10482605 and rs6198 polymorphisms in admixed populations and explore their impact on GRα and GRβ expression, to determine their functional role in glucocorticoid resistance. Although we did not find any methylation difference in the investigated CpG sites, our findings do not rule out this epigenetic mechanism as a regulator of *NR3C1* expression in other tissues or of other genes, since lower familial warmth in infancy was independently associated with MetS susceptibility in the Mennonite population.

## Figures and Tables

**Figure 1 genes-14-01805-f001:**
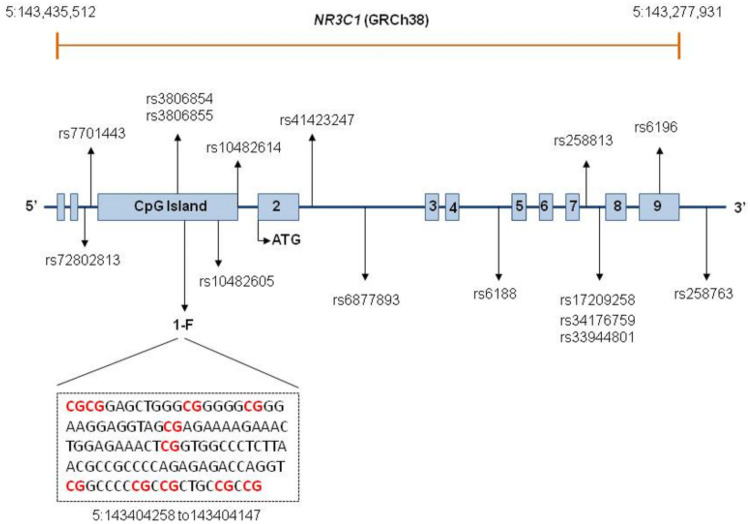
*NR3C1* structure and location of the SNPs comprising the investigated haplotypes and CpGs.

**Figure 2 genes-14-01805-f002:**
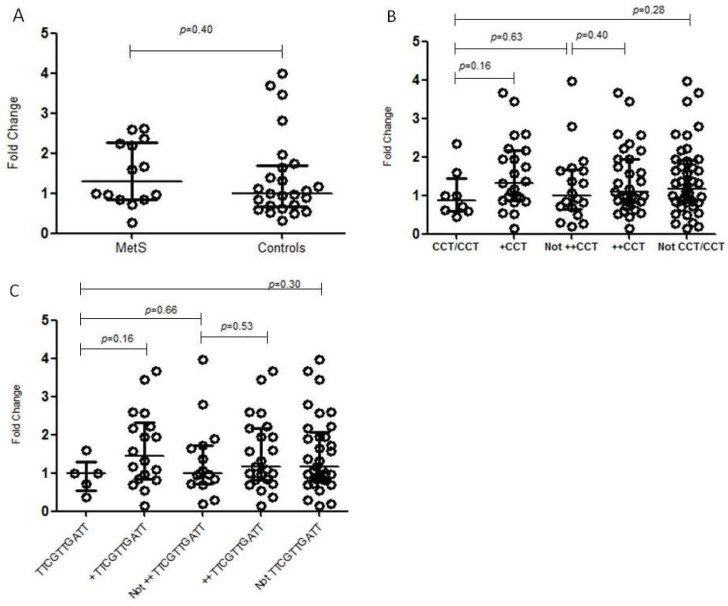
*NR3C1* mRNA expression in the buffy coat, according to different genotypes. The horizontal bars in the clusters indicate the median; *p*-values were calculated by Mann–Whitney’s test: (**A**) association between the expression of *NR3C1* mRNA and MetS; (**B**) association between the expression of *NR3C1* mRNA and *NR3C1 CCT* haplotype. +*CCT*—heterozygote individuals; ++*CCT*—homozygote or heterozygote individuals; (**C**) association between the expression of *NR3C1* mRNA and *NR3C1 TTCGTTGATT* haplotype. *TTCGTTGATT*—homozygotes; +*TTCGTTGATT*—heterozygote individuals; ++*TTCGTTGATT*—homozygote or heterozygote individuals.

**Table 1 genes-14-01805-t001:** Diagnostic criteria for MetS.

Components		Criteria
Waist circumference	Male	≥94 cm
	Female	≥80 cm
Triglycerides ^a^		≥150 mg/dL
HDL cholesterol ^a^	Male	<40 mg/dL
	Female	<50 mg/dL
Blood pressure ^a,b^	Systolic and/or	≥130 mm Hg
	Diastolic	≥85 mm Hg
Type 2 diabetes mellitus ^a^		Self-reported medical diagnostic

^a^ Specific drug treatment is an alternative indicator. ^b^ Diagnosis of hypertension is an alternative indicator. Modified from Alberti et al. [10].

**Table 2 genes-14-01805-t002:** Distribution of characteristics in Mennonite population samples.

	Controls				Individuals with MetS			
	Total	CWB	CWI	CON	Total	CWB	CWI	CON
N	237	91	65	81	112	35	19	58
Gender (M/F)	91/146	35/56	25/40	31/50	57/55	21/14	11/8	25/33
Average age	44.70	41.50	40.68	51.51	59.85	57.00	54.13	63.41
(min–max) (years)	(12.30–92.07)	(12.30–89.00)	(14.42–92.07)	(16.41–83.37)	(19.96–85.91)	(29.00–82.30)	(19.96–77.77)	(36.56–85.91)
Prevalence	-	-	-	-	32.09	27.77	22.62	41.72
	%	%	%	%	%	%	%	%
Waist circumference: male ≥ 94 cm/female ≥ 80 cm	78.51	78.73	69.38	85.71	100	100	100	100
Triglycerides ≥ 150 mg/dL	23.21	31.19	20.54	16.32	90.99	91.17	100	87.93
HDL cholesterol: male < 40 mg/dL/female < 50 mg/dL	8.92	9.17	4.10	12.24	59.80	45.71	42.10	50.00
Blood pressure Systolic ≥ 130 mm Hg/Diastolic ≥ 85 mm Hg	22.46	19.13	14.66	33.72	81.98	77.14	68.42	89.47
Type 2 DM	1.09	2.72	0	0	20.72	14.70	21.05	24.13
Moderate/vigorous physical activity	70.28	77.41	65.33	66.66	56.63	51.42	60.00	56.66
Family environment during childhood								
Cold	10.12	10.86	2.73	15.85	20.00	25.71	10.00	20.00
Moderate	47.36	38.04	58.90	47.56	56.52	57.14	65.00	53.33
Warm	42.51	51.86	38.35	36.58	23.47	17.14	25.00	26.66

DM—diabetes mellitus; F—female; M—male; N—number of individuals; CON—Colônia Nova, RS; CWB—Curitiba, PR; CWI—Witmarsum, PR.

**Table 3 genes-14-01805-t003:** *NR3C1* SNPs investigated through exome analysis.

Polymorphism	Position	HGVS Description	Location
rs1192533423	5:143404595	ENST00000343796.6:c.-13-3743T>G	Regulatory region
rs3806855	5:143404564	ENST00000343796.6:c.-13-3712T>G	Regulatory region
rs3806854	5:143404562	ENST00000343796.6:c.-13-3710T>C	Regulatory region
rs5871845	5:143404390	ENST00000343796.6:c.-13-3533_-13-3532insC	Regulatory region
rs10482605	5:143403956	ENST00000343796.6:c.-13-3104T>C	Regulatory region
rs10482606	5: 143403703	ENST00000343796.6:c.-13-2851T>C	Regulatory region
rs571795102	5:143403515	ENST00000343796.6:c.-13-2663A>G	Regulatory region
rs10482614	5:143402837	ENST00000343796.6:c.-13-1985G>A	Regulatory region
rs192978343	5:143402635	ENST00000343796.6:c.-13-1783T>G	Regulatory region
rs6189	5:143400774	ENST00000343796.6:c.66G>A	Exon 2
rs6190	5:143400772	ENST00000343796.6:c.68G>A	Exon 2
rs56149945	5:143399752	ENST00000343796.6:c.1088A>G	Exon 2
rs6188	5:143300779	ENST00000343796.6:c.1469-16G>T	Intron 4
rs761295829	5:143300547	ENST00000343796.6:c.1685C>T	Exon 5
rs258813	5:143295125	ENST00000343796.6:c.2023+335C>T	Intron 7
rs33944801	5:143294377	ENST00000343796.6:c.2023+1083G>C	Intron 7
rs34176759	5:143294375	ENST00000343796.6:c.2023+1085del	Intron 7
rs17209258	5:143293832	ENST00000343796.6:c.2023+1628T>C	Intron 7
rs926407137	5:143282772	ENST00000343796.6:c.2024-47G>A	Intron 7
rs6196	5:143281925	ENST00000343796.6:c.2298T>C	Exon 9

**Table 4 genes-14-01805-t004:** Association test of *NR3C1* polymorphisms with MetS.

Polymorphism	Reference Allele	Controls	MetS Individuals	Dominant Model		Additive Model		Recessive Model	
		N/N Total (Frequency)	N/N Total (Frequency)	OR [CI 95%]	*p*corr	OR [CI 95%]	*p*corr	OR [CI 95%]	*p*corr
rs72802813 C>t	*T*	60/326 (0.18)	30/140 (0.21)	1.22 [0.65–2.26]	0.60	1.17 [0.68–2.02]	0.61	1.08 [0.18–6.44]	
rs7701443 T>c	*C*	137/324 (0.42)	54/134 (0.40)	0.70 [0.36–1.33]	0.23	0.81 [0.53–1.25]	0.38	0.85 [0.38–1.86]	0.85
rs1192533423 T>g	*G*	11/276 (0.039)	5/148 (0.033)	1.08 [0.32–3.71]	0.91	1.08 [0.32–3.71]	0.91	n.d.	n.d.
rs3806855 T>g	*G*	58/274 (0.21)	28/148 (0.18)	0.67 [0.33–1.36]	0.28	0.68 [0.39–1.21]	0.19	0.42 [0.09–1.91]	0.25
rs3806854 T>c	*C*	58/274 (0.21)	28/148 (0.18)	0.67 [0.33–1.36]	0.28	0.68 [0.39–1.21]	0.19	0.42 [0.09–1.91]	0.25
rs5871845 G>gc	*GC*	11/276 (0.03)	10/148 (0.06)	1.49 [0.53–4.22]	0.46	1.49 [0.53–4.22]	0.46	n.d.	n.d.
rs10482605 T>c	*C*	54/276 (0.19)	38/148 (0.25)	1.19 [0.61–2.31]	0.62	1.42 [0.83–2.43]	0.2	4.74 [1.11–20.29]	**0.024**
rs10482606 T>c	*C*	2/276 (0.006)	1/148 (0.007)	5.44 [0.34–87.96]	0.11	5.44 [0.34–87.96]	0.11	n.d.	n.d.
rs571795102 A>g	*G*	6/276 (0.02)	6/148 (0.04)	1.63 [0.42–6.25]	0.48	1.63 [0.42–6.25]	0.48	n.d.	n.d.
rs10482614 G>a	*A*	60/276 (0.21)	30/148 (0.20)	0.74 [0.37–1.47]	0.37	0.72 [0.41–1.25]	0.22	0.39 [0.09–1.75]	0.23
rs41423247 C>g	*G*	123/324 (0.37)	43/140 (0.30)	0.65 [0.35–1.20]	0.22	0.82 [0.52–1.29]	0.42	1.14 [0.47–2.74]	0.70
rs6877893 C>t	*T*	143/322 (0.44)	66/140 (0.47)	0.95 [0.49–1.84]	0.85	1.03 [0.68–1.56]	0.85	1.17 [0.57–2.39]	0.70
rs192978343 T>g	*G*	3/276 (0.01)	1/148 (0.006)	3.95 [0.3–52.72]	0.14	3.95 [0.3–52.72]	0.14	n.d.	n.d.
rs6189 G>a	*A*	15/276 (0.05)	7/148 (0.04)	0.83 [0.29–2.37]	0.76	0.83 [0.29–2.37]	0.76	n.d.	n.d.
rs6190 G>a	*A*	15/276 (0.05)	7/148 (0.04)	0.83 [0.29–2.37]	0.76	0.83 [0.29–2.37]	0.76	n.d.	n.d.
rs56149945 A>g	*G*	10/276(0.036)	5/148 (0.033)	0.94 [0.28–3.15]	0.78	0.94 [0.28–3.15]	0.78	n.d.	n.d.
rs6188 G>t	*T*	114/276 (0.41)	68/148 (0.45)	1.32 [0.65–2.66]	0.45	1.02 [0.65–1.6]	0.92	0.74 [0.33–1.67]	0.46
rs761295829 C>t	*T*	1/276 (0.003)	1/148 (0.006)	0.71 [0.04–12.69]	0.87	0.71 [0.04–12.69]	0.87	n.d.	n.d.
rs258813 C>t	*T*	114/276 (0.41)	68/148 (0.45)	1.32 [0.65–2.66]	0.45	1.02 [0.65–1.6]	0.92	0.74 [0.33–1.67]	0.46
rs33944801 G>c	*C*	32/272 (0.11)	11/146 (0.07)	0.6 [0.24–1.47]	0.27	0.68 [0.31–1.52]	0.32	1.16 [0.11–11.66]	0.94
rs34176759 A>t	*T*	32/272 (0.11)	11/146 (0.07)	0.6 [0.24–1.47]	0.27	0.68 [0.31–1.52]	0.32	1.16 [0.11–11.66]	0.94
rs17209258 T>c	*C*	54/276 (0.19)	21/148 (0.14)	0.71 [0.34–1.44]	0.34	0.79 [0.42–1.47]	0.45	1.19 [0.2–6.98]	0.88
rs926407137 G>a	*A*	3/276 (0.01)	1/148 (0.006)	0.45 [0.04–4.69]	0.33	0.45 [0.04–4.69]	0.33	n.d.	n.d.
rs6196 T>c	*C*	60/276 (0.21)	30/148 (0.20)	0.74 [0.37–1.47]	0.37	0.72 [0.41–1.25]	0.22	0.39 [0.09–1.75]	0.23
rs258763 a>T	*A*	142/326 (0.43)	58/138 (0.42)	0.85 [0.45–1.63]	0.70	0.87 [0.57–1.33]	0.47	0.80 [0.37–1.72]	0.75

The minor allele for the Mennonite population is shown in lowercase. *p*-value was corrected to the Monte Carlo permutation method and independent variables. In bold—significant *p*corr value; OR—odds ratio; CI—confidence interval; N—number of alleles; n.d.—no data to perform statistical analysis.

**Table 5 genes-14-01805-t005:** Frequencies of *NR3C1* haplotypes and association assay.

Haplotype			Dominant Model			Additive Model			Recessive Model		
	Controls N/N Total	Patients N/N Total	OR	CI 95%	*p*corr	OR	CI 95%	*p*corr	OR	CI 95%	*p*corr
rs41423247/rs6877893/rs258763											
*Cta*	141/324	58/138	0.81	0.43–1.53	0.532	0.85	0.56–1.30	0.468	0.79	0.37–1.71	0.565
*gCT*	122/324	41/138	0.61	0.33–1.13	0.117	0.76	0.48–1.20	0.251	0.99	0.40–2.45	0.986
*CCT*	56/324	33/138	1.09	0.58–2.04	0.782	1.36	0.81–2.26	0.236	6.02	1.41–25.62	**0.015 (0.030)**
rs72802813/rs7701443											
*Cc*	135/318	54/134	0.69.	0.36–1.32	0.271	0.81	0.52–1.24	0.613	0.84	0.38–1.84	0.669
*CT*	124/318	52/134	1.04	0.54–1.99	0.903	1.21	0.75–1.94	0.339	1.81	0.75–4.36	0.183
*tT*	58/322	28/136	1.17	0.62–2.19	0.613	1.25	0.68–2.28	0.429	0.61	0.05–6.55	0.688
*TTTGGCGATT **	101/256	52/132	1.18	0.60–2.31	0.615	1.07	0.67–1.68	0.770	0.95	0.39–2.31	0.918
*TTcGttGATT **	50/256	36/132	1.18	0.61–2.30	0.614	1.47	0.86–2.52	0.155	4.74	1.10–20.28	**0.036 (0.048)**
*gcTattGATc **	52/256	23/132	0.67	0.33–1.37	0.282	0.68	0.38–1.21	0.195	0.41	0.09–1.91	0.262

The minor allele for the Mennonite population is shown in lowercase. *p*-value was corrected to the Benjamini and Hochberg method and independent variables. * Haplotype represented ten investigated *NR3C1* SNPs (rs3806855, rs3806854, rs10482605, rs10482614, rs6188, rs258813, rs33944801, rs34176759, rs17209258, and rs6196). In bold—significant *p*corr; OR—odds ratio; CI—confidence interval; N—number of alleles.

**Table 6 genes-14-01805-t006:** *NR3C1* CpG methylation levels.

CpG Site	Unmethylated n (%)		n		*p*-Value *
	Controls	MetS Individuals	Controls	MetS Individuals	
35	55 (41)	23 (39)	131	58	0.82
36	71 (54)	26 (44)	131	58	0.52
37	95 (72)	40 (68)	131	58	0.76
38	115 (90)	56 (96)	127	58	0.13
39	119 (90)	55 (94)	131	58	0.34
40	65 (05)	28 (45)	130	61	0.89
43	110 (86)	52 (86)	127	60	0.94
44	100 (78)	47 (78)	127	60	0.89
45	117 (92)	53 (88)	127	60	0.38
46	124 (97)	59 (98)	127	60	0.75
47	127 (99)	58 (98)	128	59	0.58
Median overall methylation	-	-	111	51	0.76

CpG nomenclature according to Palma-Gudiel et al. [20]. * Mann–Whitney test; n—number of individuals.

## Data Availability

The data presented in this study are available upon request from the corresponding author. The data are not publicly available due to privacy restrictions given by the General Law of Data Protection (LGPD) in Brazil.

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
