# Peer review of "Glucocorticoid Receptor Gene (NR3C1) Polymorphisms and Metabolic Syndrome: Insights from the Mennonite Population"

_genes, 2023, doi:10.3390/genes14091805_

Round 1

Reviewer 1 Report

  1. The abstract should contain a clearly formulated aim of the study.  
  2. How was familiar warmth in infancy and physical activity defined in the study? Please explain in the manuscript.
  3. Can an institution be among the authors of the paper? If not, please remove Regeneration Genetics Center from the list of authors.
  4. Please provide a table with characteristics of studied groups, including blood parameters,  physical activity levels and family environment during childhood.
  5. Figure 1 – are all the investigated SNPs depicted in the figure? One of polymorphisms from intron 7 that is present in table 3 is missing from figure 1, polymorphism localized in exon 5 is also missing.

Reviewer 2 Report

Manuscript review response: “Glucocorticoid Receptor Gene (NR3C1) Polymorphisms and Metabolic Syndrome: Insights From The Mennonite Population”.

Manuscript ID: genes-2581565

Comments

It is an article that could provide relevant information about the polymorphisms of NR3C associated to Metabolic Syndrome in a special population, but is necessary there are some changes, add some inscriptions and information.

Line 67-68 What are the parameters associated of MetS? you could add it in this section.

Line 79 What transcription factors are associated with expression of inflammatory genes? Add the information please.

Line 102 What are stressors factors attributed to DNA methylation alterations in childhood? Add some stressors factors.

Line 113-116 it is important to remark the aim of this study between Glucocorticoid Receptor Gene (NR3C1) Polymorphisms and metabolic syndrome. your tittle is "Glucocorticoid Receptor Gene (NR3C1) Polymorphisms and Metabolic Syndrome: Insights from The Mennonite Population". could you add why is important analyser these polymorphisms for this work? could you add information about these polymorphisms in other ethnicities or population?

Line 210-215 In figure 1 only the letter F is described, you could add the corresponding letters in each number (in the figure). The letter G is missing in the description of figure 1.

Line 216 the subtitle says “mRNA Quantification” you could add RNA extraction, cDNA synthesis and RT-qPCR.

Line 217-227 what was equipment the RT-PCR used? what were the integrity and purity of RNA? You could add if was TaqMan Probe or Sybr green. Could you add a primers sequence used in a table?

Line 266-269 did you used to mean with standard deviations?

Line 352 say “methylation or expression levels” change by “methylation or gene expression levels”.

Moderate editing of English language required
